# Strong lowering of ionization energy of metallic clusters by organic ligands without changing shell filling

Vikas Chauhan[1], Arthur C. Reber [1] & Shiv N. Khanna [1]

Alkali atoms have unusually low ionization energies because their electronic structures have an excess electron beyond that of a filled electronic shell. Quantum states in metallic clusters are grouped into shells similar to those in atoms, and clusters with an excess electron beyond a closed electronic may also exhibit alkali character. This approach based on shell-filling is the way alkali species are formed as explained by the periodic table. We demonstrate that the ionization energy of metallic clusters with both filled and unfilled electronic shells can be substantially lowered by attaching ligands. The ligands form charge transfer complexes where the electronic spectrum is lifted via crystal field like effect. We demonstrate that the effect works for the weakly bound ligand, *N*-ethyl-2-pyrrolidone (EP = $C_6H_{11}NO$), and that the effect leads to a dramatic lowering of the ionization energy independent of the shell occupancy of the cluster.

[1] Department of Physics, Virginia Commonwealth University, Richmond, VA 23284-2000, USA. Correspondence and requests for materials should be addressed to S.N.K. (email: Snkhanna@vcu.edu)

onization energy and electron affinity are fundamental electronic characteristics of elements and are generally governed by the electron count. For atoms, these features govern their tendency to achieve a closed shell status either by donating or receiving electrons as they combine with other elements to form, e.g., ionic solids. While these characteristics are inherent to the elements, metallic clusters also share similar features. Quantum confinement of electron gas in metallic clusters leads to a grouping of the electronic states into shells as in the atoms[1–6]. These shells order as $1S^2$, $1P^6$, $1D^{10}$, $2S^2$, $1F^{14}$, $2P^6$, $1G^{18}$… where the uppercase letters are used to distinguish them from atoms that have a different combination of the principal and angular momentum quantum numbers. Extensive experiments on the mass abundances, ionization energies, electron affinities, and reactivity of metal clusters provide evidence that clusters containing 2, 8, 18, 20, 34, 40… electrons leading to filled electronic shells are more abundant than the other sizes. Furthermore, neutral clusters with filled electronic shells exhibit high ionization energy and low electron affinity, while the clusters that have one electron past the filled shell or are deficient in an electron to complete electronic shell exhibit lower ionization energy or higher electron affinity compared to neighbors. This analogy between metallic clusters and atoms has led to the concept that stable clusters with a well-defined valence can be classified as superatoms forming a 3rd-dimension of the periodic table. Over the past two decades, numerous superatoms resembling the alkali, halogen, inert, and magnetic atoms have been proposed[7–11]. For example, an $Al_{13}$ with 39 valence electrons occupying $1S^21P^61D^{10}2S^21F^{14}2P^5$ shells requires one electron to fill the last shell. Our previous study showed that it has a very high electronic affinity of 3.6 eV, reminiscent of a chlorine atom. On the other hand, an $Al_{13}^-$ with filled electronic shells is highly stable species as shown by its resistance to reactivity with oxygen even though bulk Al readily oxidizes. The identification of stable clusters with a well-defined valence has led to another promising direction in nanoscience namely synthesizing nano-scale materials with superatoms as building blocks[12]. While such solids have been synthesized, in many cases, the reactive nature of metallic clusters has prevented stability of such solids as the clusters coalesce when put together. One way to circumvent this problem is to use ligands around the metallic cores that avoid a direct interaction between the building blocks[13,14]. In addition to passivating surfaces, the ligands can also alter the valence electronic count of the cluster offering a viable path to stabilize clusters. These two aspects have been highly effective in the stability of numerous cluster assemblies synthesized over the past few years[15,16]. For example, numerous assemblies based on pure gold or mixed clusters ligated with thiols such as $Au_{102}(SR)_{44}$[17–19] have been synthesized. The stability these ligated clusters is generally rationalized within the superatomic framework. For example, in $Au_{102}(SR)_{44}$, the outer 44 gold atoms are linked to thiols making covalent bonds, while the 58 atoms within the core provide of 58 electrons which leads to a closed electronic shell. In fact, nearly all the stable gold–thiol clusters are demonstrated as superatoms with closed electronic shells and large HOMO-LUMO gaps by Häkkinen and co-workers[19].

In this paper, we propose an alternative strategy to design the superatomic clusters that can be transformed into donors or acceptors of multiple electrons independent of whether the cluster has an open or closed electronic shells. These clusters resemble multiple valence superatoms as they can combine with counter ions of different valence to form stable cluster assemblies or simply can be used as donors/acceptors for a variety of applications such as multiple electron donors for semiconductors[20–22]. We accomplish the above transformation by attaching the suitable ligands to the metallic cores that form charge transfer complexes. Unlike thiol ligands, these ligands leave the effective valence count of the metallic core intact, though the exchange of local charge does take place between the ligands and metallic core. We demonstrate this phenomena by successive attachment of three N-ethyl-2-pyrrolidone (EP = $C_6H_{11}NO$) ligands to a bare $Al_{13}$ and doped $MAl_{12}$ (M = B, C, Si, and P) clusters containing 39, 40, and 41 valence electrons. An EP ligand is a monomer unit of Poly(N-vinyl-2-pyrrolidone) that has been shown to stabilize gold clusters and an exchange charge with metal core in the reported experimental and theoretical studies by Tsunoyama et al.[23,24]. The Al based clusters are chosen because of their metallic nature and the applicability of the confined nearly free electron gas model to describe their electronic structure. We show that the local charge donated to the metallic core primarily lifts the electronic spectrum to lower the ionization energy. This process is analogous to the shift in the work function of a surface based on a dipole moment, although additional effect are found to enhance the lowering of the ionization potential beyond that of a simple electrostatics argument[25–29]. Another crucial result of our study is that successive ligation not only lowers the ionization energy but also that the amount of lowering of the ionization energy is comparable irrespective of the filling of the valence shell. The observed ligation effect is even more spectacular for the clusters with highest occupied molecular orbital (HOMO) that contains multiple electrons. Furthermore, this effect allows for the formation of effective multiple electron donors as the reduction in the 2nd ionization energy of the bare species can be even more than that of in the 1st ionization energy.

## Results

**Analysis of $MAl_{12}$ clusters, M = B, C, Si, and P**. Our investigation starts with the structural analysis of bare $Al_{13}^{0/-}$ and doped $MAl_{12}$ (M = B, C, Si, and P) clusters. Figure 1 shows the optimized ground state structures of the clusters along with selected Al–Al bond lengths. All the geometries were optimized without any symmetry constraint. We have considered only endohedral doped Al icosahedral clusters since previous theoretical studies have reported that the endohedral doping is energetically more favorable than the exohedral doping[30,31]. The $Al_{13}$ and $BAl_{12}$ both have 39 valence electrons with electronic configuration of $1S^21P^61D^{10}2S^21F^{14}2P^5$. Since the icosahedral clusters have a degenerate 2P state that is partially filled, the clusters undergo a Jahn–Teller distortion from the symmetric shape as seen from a variation in the bond lengths. For example, the $Al_s$–$Al_s$ bond lengths in $Al_{13}$ vary from 2.76 to 2.96 Å while it varies from 2.65 to 2.73 Å in $BAl_{12}$. Here subscript s stands for the Al atoms on the surface. For $BAl_{12}$, one expects a contraction in $Al_s$–$Al_s$ distances as well as in the average bond length from the central site to $Al_s$ sites due to the smaller radius of a B atom. The average bond length from central site to $Al_s$ sites in $Al_{13}$ indeed decreases from 2.69 to 2.56 Å in $BAl_{12}$. To achieve closed electronic shell of 40 e$^-$($1S^21P^61D^{10}2S^21F^{14}2P^6$), the central Al site was replaced with a C or a Si atom both of which contribute 4 valence e$^-$'s. The closed electronic shell stabilizes the symmetrical icosahedral structure of $CAl_{12}$ and $SiAl_{12}$ as revealed by their $Al_s$-$Al_s$ bond lengths (2.68 and 2.79 Å) shown in Fig. 1. The bond lengths in $CAl_{12}$ and $SiAl_{12}$ cages are slightly smaller than in iso-electronic $Al_{13}^-$ due to the charged state. In order to create an analog of an alkali atom, the central Al atom was then replaced by a P atom that results in a valence pool of 41 e$^-$'s. Since the extra electron is in a degenerate state of the symmetric cluster, a Jahn–Teller distortion with different surface bonds ($Al_s$–$Al_s$ = 2.73–2.80 Å) is observed.

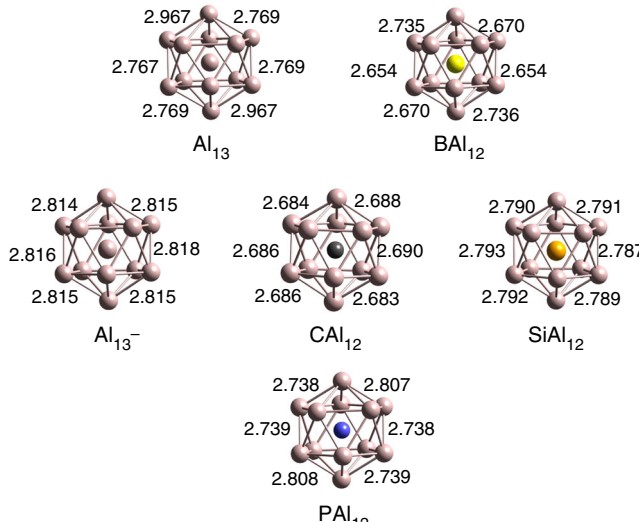

**Fig. 1** Ground state structures of $Al_{13}^{0/-}$ and $MAl_{12}$ (M = B, C, Si, and P) clusters. The Al–Al bond lengths are also shown and given in Å

Here, we are primarily interested in assessing the effect of shell filling on the electronic properties. Therefore, the adiabatic ionization energies (AIEs) of $Al_{13}$ and $MAl_{12}$ (M = B, C, Si, and P) were calculated by taking the energy difference between the ground state of the neutral and cationic species. The AIE values are plotted in Fig. 2 and given in Supplementary Table 1. It is interesting that the clusters with same valence electron count (despite a different composition) have comparable AIEs. The $CAl_{12}$ and $SiAl_{12}$ due to the electronic shell closure are marked by their high AIE of 6.63 and 6.73 eV respectively while $Al_{13}$ has a AIE of 6.13 eV slightly less than that of $BAl_{12}$ (6.64 eV) as $Al_{13}^+$ is stabilized in the oblate structure than the icosahedral structure. Note that AIEs show only a small increment as one passes from 39 $e^-$ ($Al_{13}$ and $BAl_{12}$) to 40 $e^-$ systems ($CAl_{12}$ and $SiAl_{12}$) because of the same 2 P outer valence electronic shell, then a reduction of 1.5 eV in the AIE is observed as one approaches to 41 $e^-$ system ($PAl_{12}$). This sharp drop in the AIE represents the shell effect similar to atoms. The effect is, however, sizable only for one electron past the closed electronic shell.

**Analysis of ligated clusters**. At this point, we wanted to investigate if it is possible to go beyond the shell effect namely, is it possible to create alkali-like species without changing the valence count? More importantly, can one form super donors that can donate multiple electrons without substantial increase in the successive ionization energy? We will now show that this can be accomplished by attaching ligands that form charge transfer complexes. An EP ligand with donor characteristics looked promising. To explore this exciting possibility, we first examined the ground state structures of the $Al_{13}(EP)_n$ and $MAl_{12}(EP)_n$ (n = 1–3) clusters. Various positions of the ligands were tried to find the ground state structures. Figure 3 shows the ground state structures of the ligated clusters. While the 1st EP ligand binds to top Al site of the metallic core, 2nd EP ligand always prefers the *meta* position rather than an *ortho* or *para* position in the ground state. We find that the *para* position is destabilized by the ligand inducing an electron donor site on the opposite side of the clusters with closed electronic shells such as $CAl_{12}$ and $SiAl_{12}$[13]. For the case of open shell $Al_{13}$, and $BAl_{12}$, the *para* position is only slightly less stable because the induced active site is half-filled. The *ortho* position is close in energy to the *meta* position in all cases, but is destabilized due to steric interactions. The *ortho*, *meta*, and *para* positions relative to 1st ligand are shown in

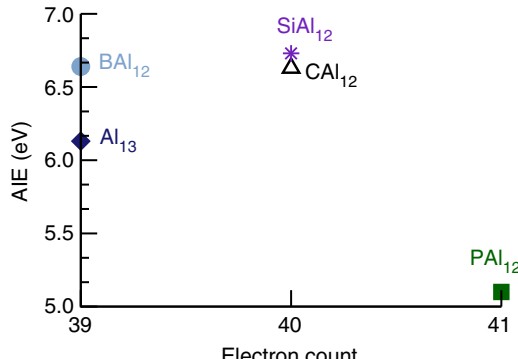

**Fig. 2** Electronic properties of ligand-free Al and doped Al clusters. Adiabatic ionization energies of bare $Al_{13}$, $MAl_{12}$ (M = B, C, Si, and P) clusters. All the values are given in eV

Supplementary Fig. 1. In $Al_{13}(EP)_3$, $SiAl_{12}(EP)_3$, and $PAl_{12}(EP)_3$, the 3rd ligand stabilized in an *ortho* position, while it is the *meta* position for $BAl_{12}(EP)_3$ and $CAl_{12}(EP)_3$. Total energies of the optimized structures with various positioning of the ligands are given in Supplementary Figs. 1–3. As expected, the successive attachment of ligands induces the distortion in the symmetric $CAl_{12}$ and $SiAl_{12}$ clusters, for instance, $Al_s$–$Al_s$ bond length varies from 2.68 to 2.80 Å in $CAl_{12}(EP)_n$ (n = 1–3). Similar variation in $Al_s$–$Al_s$ bond length is observed for other ligated $MAl_{12}$ clusters. Interestingly, the average bond length from the central site to $Al_s$ site is nearly unchanged during ligation indicating that the bonding characteristics within the clusters were largely unperturbed. Knowing the donor characteristic of an EP ligand, we determined the gross charge on the metallic core in the ground states via Mulliken population analysis. We have given the calculated values in Supplementary Table 2. Figure 4a shows the charge accumulated $\Delta Q_{n-(n-1)}$ (relative to corresponding precursors) on the metallic core during successive ligation. The $BAl_{12}$ and $Al_{13}$ core gain 0.40 and 0.39 $e^-$ respectively from the 1st EP ligand while it is 0.38 and 0.36 $e^-$ for $CAl_{12}$ and $SiAl_{12}$, respectively. On the other hand, charge transfer from 1st ligand to $PAl_{12}$ is found to be 0.35 $e^-$. One can see that $\Delta Q_{n-(n-1)}$ does not decrease with increasing number of ligands. In fact, 3rd EP ligand donates more charges than the 2nd EP ligand as shown in Fig. 4a. However, nearly equal amount of charge on the metallic core with a given number of ligand implies the non-sensitivity of charge transfer to the electronic character of the bare clusters. Since EP ligands donate the charge to core, it will be interesting to see how strongly the ligands are bound to the cluster. We therefore calculated the successive ligand-binding energy using Eq (1).

$$BE = E\left(MAl_{12}(EP)_{n-1}\right) + E(EP) - E\left(MAl_{12}(EP)_n\right), \qquad (1)$$

where n = 1–3 and M = Al, B, C, Si and P. Here E presents the total energy of the system. Figure 4b shows the variation in the calculated BE, while the numerical values are given in Supplementary Table 3. The BE is found to decrease with increasing number of ligands due to charge accumulation on the precursors, though the 1st EP ligand binds more strongly to $Al_{13}$ and $BAl_{12}$ than others. This is because of their electron deficiency to complete the electronic shell. The BE of 2nd EP ligand turns out to be around 0.7 eV irrespective of the clusters consistent with the amount of the gross charge on the metallic core. Similar results are obtained with 3rd EP ligand.

Although the BE of ligands turns out to be small, we find that the ligation dramatically influences the electronic properties. To demonstrate this, the variation in the AIEs of ligated Al clusters are shown in Fig. 5a. The AIEs in all the cases continuously

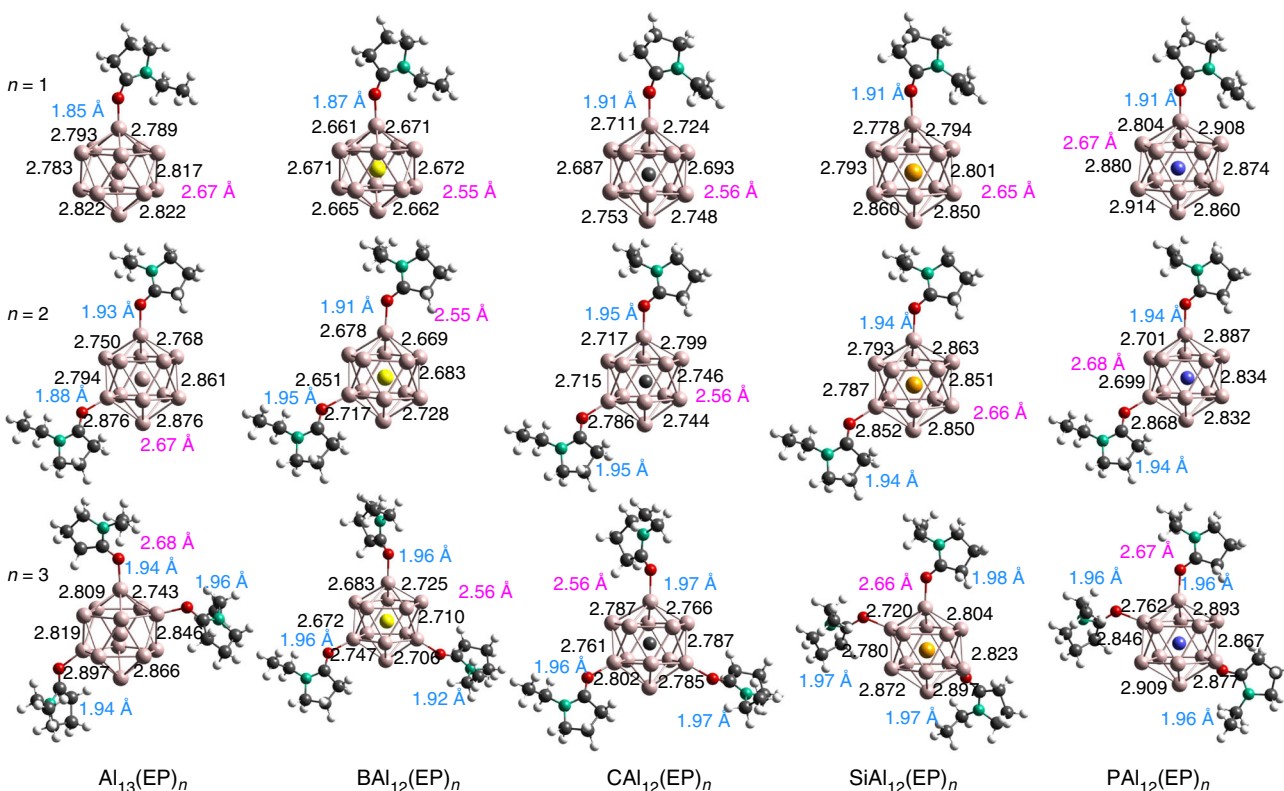

**Fig. 3** Ground state structures of neutral $Al_{13}(EP)_n$ and $MAl_{12}(EP)_n$ (M = B, C, Si, and P) clusters where n = 1–3. The Al–Al bond lengths are also shown and given in Å. The average bond lengths of interior central atom to surface Al atoms are given in magenta text, while the Al–O bond lengths are given in light blue text

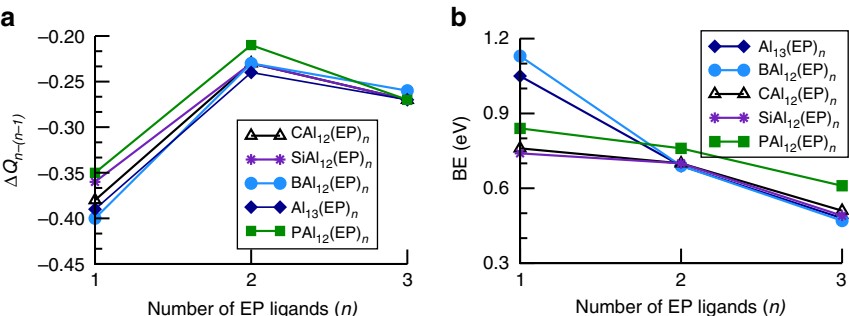

**Fig. 4** The effect of ligands on Al and doped Al clusters. **a** Accumulated charge and **b** Ligand-binding energies in $Al_{13}(EP)_n$ and $MAl_{12}(EP)_n$ (M = B, C, Si, and P) clusters where n = 1–3

decline with the successive addition of EP ligand, for instance, 1st EP ligands drop the AIE of $CAl_{12}$ by 1.06 eV while it is 2.47 eV with 3 EP ligands. Similarly, $Al_{13}(EP)_3$ turns out to have AIE of 3.51 eV smaller than that of $Al_{13}$. All the values of AIEs are given in Supplementary Table 1. The identical trend in the AIEs indicate the emergence of alkali-like character, though the underlying cause of this captivating phenomena is not clear. What is even more intriguing is that not only the decrease in the AIEs systematically increases with the number of ligands but also that the magnitude turns out to be almost identical irrespective of the initial clusters. At first, one might guess that even though ligands do not attached to clusters very strongly, yet ligands alter the effective valence electron count. However, this hypothesis turns out to be incorrect. According to this hypothesis, the AIE of $Al_{13}$ should increase since the charge donation of around $0.90e^-$ in $Al_{13}(EP)_3$ would make it closer to have a filled shell. However,

AIE of $Al_{13}$ gets reduced to 3.51 eV from 6.13 eV as shown in Fig. 5a. This interesting scenario is even more puzzling if one looks at the multiple ionization energies of the $CAl_{12}$ and $Al_{13}$ clusters. Figure 5b compares the 1st and 2nd AIE of the $CAl_{12}$ and $CAl_{12}(EP)_3$ clusters. The 2nd AIE of $CAl_{12}$ and $CAl_{12}(EP)_3$ are found to be 9.97 and 6.84 eV, respectively. Note that the 2nd AIE is reduced by 3.13 eV more than (2.62 eV) the 1st AIE. Moreover, the 2nd AIE of $CAl_{12}(EP)_3$ is less than the 1st ionization energy of most non-alkali atoms. On the other hand, 2nd AIE of $Al_{13}(EP)_3$ is found to be 7.21 eV compared to the corresponding value of 9.60 eV for an $Al_{13}$. All these results imply that decrease in the AIE is probably not associated with the filling of the electronic shell.

**Analysis of electronic structure**. To gain better insight into the nature of decrease, we examined the one-electron energy levels of

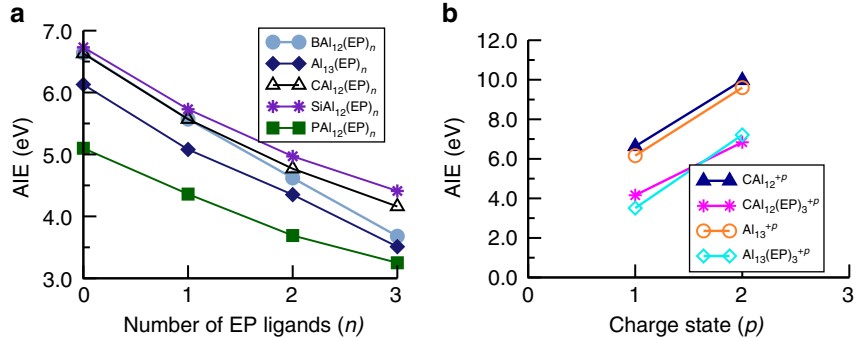

**Fig. 5** Electronic properties of ligated Al clusters. **a** Adiabatic ionization energies of bare $Al_{13}(EP)_n$, $MAl_{12}(EP)_n$ (M = B, C, Si, and P; n = 0–3) and **b** First and second adiabatic ionization energies of bare $CAl_{12}$, $CAl_{12}(EP)_3$, $Al_{13}$, and $Al_{13}(EP)_3$ clusters. All the values are given in eV

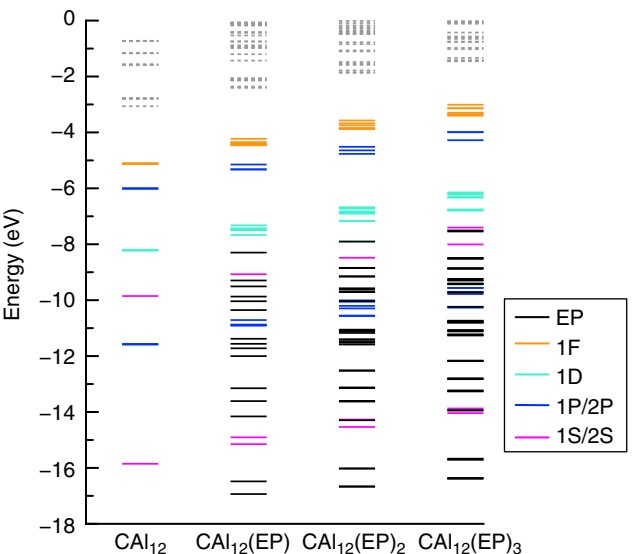

**Fig. 6** One-electron energy levels of $CAl_{12}(EP)_n$ (n = 0–3) clusters. The occupied and unoccupied states are represented by solid and dashed lines. The levels are also marked with their angular character

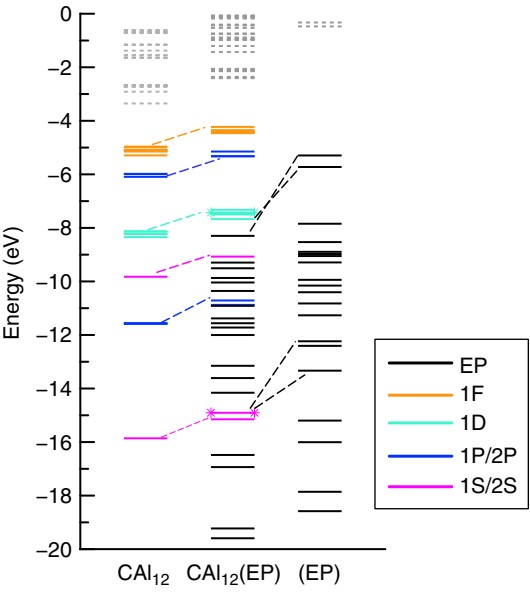

**Fig. 7** Fragment analysis of $CAl_{12}(EP)$ with $CAl_{12}$ and EP fragments. The occupied and unoccupied states are represented by solid and dashed lines. States marked with star symbol present the anti-bonding states

$CAl_{12}(EP)_n$ (n = 0–3) clusters. Figure 6 shows the one-electron levels of $CAl_{12}(EP)_n$ (n = 0–3) where all the clusters have singlet ground states. We noticed two major effects in the electronic spectrum. First, degeneracy in the superatomic states of $CAl_{12}$ is lifted due to geometric distortion induced by the ligation. Second, the addition of ligands continuously lifts the superatomic states towards higher energy as shown by the absolute energy of the HOMO. For example, the HOMO of $CAl_{12}(EP)_3$ turns out to be 2.10 eV higher than that of $CAl_{12}$. Since the ionization of any electronic system occurs from the HOMO, drop in the calculated AIE of $CAl_{12}(EP)_3$ is consistent with the rise of the electronic spectrum. Note that the HOMO-LUMO gap in all the cases does not undergo any appreciable change with ligation indicating that the shell structure and the associated electron count are largely unperturbed. To further confirm this, we performed a fragment analysis by taking $CAl_{12}$ and EP as two separate fragments of an optimized $CAl_{12}(EP)$ cluster and the results are shown in Fig. 7. Such an analysis can show how superatomic states of $CAl_{12}$ interact with a EP ligand. We identified the superatomic states via states delocalized over the $CAl_{12}$. One can see that none of the frontier 1F or 2P superatomic MOs near HOMO interact with the lone pair of oxygen (of EP ligand) apart from shifting to higher energy. Therefore, frontier MOs do not participate in the charge

transfer to $CAl_{12}$. On the other hand, the interaction of the 1S and one of 1D ($D_{yz}$) superatomic MOs with the lone pair of oxygen forms the fully occupied bonding and anti-bonding states as shown in Fig. 7. Therefore, interaction between deeper superatomic states and ligands facilitate the charge rearrangement via Al–O bonding. In addition, we can identify all the superatomic MOs (Supplementary Fig. 4) that form the electronic shell closure of $CAl_{12}$ in $CAl_{12}(EP)$. While this observation for a closed shell system is quite intriguing, we found similar effect for an open shell systems. Here, one-electron energy levels of $Al_{13}(EP)_n$ (n = 0–3) with open shell electronic configuration were examined. Figure 8 shows the electronic structures of $Al_{13}(EP)_n$ demonstrating an evolution analogous to that in $CAl_{12}(EP)_n$. Both the majority and minority spin states near the HOMO undergo a rise in the energy that increases with the number of ligands. Note that one of minority 2P states in $Al_{13}(EP)_n$ (n = 0–3) always remains unoccupied even in case of $Al_{13}(EP)_3$ although a charge transfer of 0.9 e⁻ takes place from three ligands. Our result is also consistent with the results reported by Watanabe et al. as they confirm the 1F angular character of last unoccupied state[32]. All these results strongly confirm the effective valence count in $CAl_{12}$ and $Al_{13}$ unchanged after ligation.

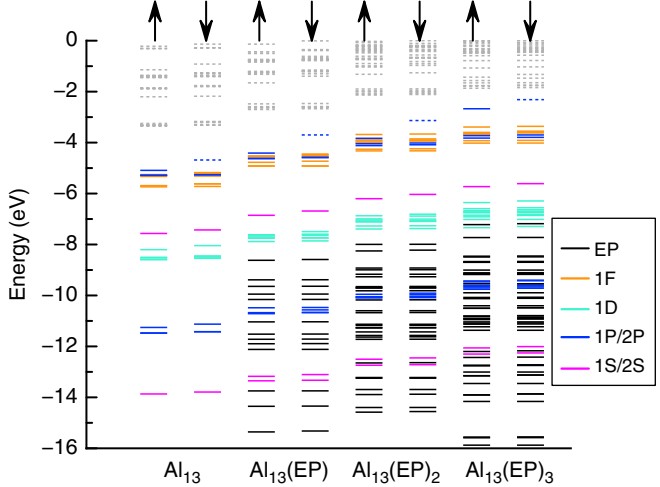

**Fig. 8** One-electron energy levels of Al$_{13}$(EP)$_n$ ($n = 0$–3) clusters. The occupied and unoccupied states are represented by solid and dashed lines. The up and down arrows indicate the majority and minority spin channels

**Analysis of charge transfer**. The above discussion brings out two important results. First, the AIE drops with successive ligation and second, this evolution is not due to change in valence count of the cluster. Hence, the microscopic origin of raising the spectrum has to be looked elsewhere. To answer this, we carried out a natural bonding orbital (NBO) analysis for CAl$_{12}$(EP)$_3$ to find out the local atomic charges. We found that there is a local rearrangement of charge reminiscent of the charge transfer complexes. Particularly, the Al sites near to oxygen gain more charge relative to others because of polarized Al–O bonds. The overall charge on CAl$_{12}$ core turns out to be $-0.39\ e^-$though much smaller than the value predicted by a Mulliken population analysis. To analyze the effect of such local charges transfer on the electronic structure, the excess charge of $0.39\ e^-$ is divided into three equal fictitious point charges of $-0.13\ e^-$ and located $0.5$ Å away from the Al sites of CAl$_{12}$ where ligands were attached. The HOMO of CAl$_{12}$ in the presence of point charges shifts to higher energy by 1.61 eV relative to that of bare CAl$_{12}$. This shift in the HOMO is only ~0.5 eV smaller than that of in the ligated cluster (2.1 eV). Therefore, a dominating effect of ligands is to create a crystal field like effect that raises the electronic spectrum while making the charge transfer complexes. Part of this effect is analogous to the change in work function caused by a dipole moment on a surface[25–28,32]. For reference, we have included the change in the work function caused by the binding of EP ligands to an Al (111) and Al (100) surface in Supplementary Fig. 5. A second related effect is that the binding energy of the ligand to the charged cluster is likely to be larger than to a neutral cluster. To separate these effects, we may use the absolute position of the HOMO in the neutral cluster to indicate the initial state electrostatic lowering of the energy levels, and the change in the adiabatic ionization energy to include both electrostatic effects, the binding enhancement of the ligand, and any other final state stabilization of the cation that may be caused by the ligands. The change in the absolute position of the HOMO of the neutral cluster is analogous to the relative position of the vacuum energy and Fermi energy in a surface. In our method, the vacuum level is 0 eV by definition, so the absolute position of the HOMO includes this electrostatic effect. For example, the addition of EP to Al$_{13}$ lowers the HOMO by 0.70 eV, while the AIE is lowered by 1.05 eV. This suggests that about 70% of the effect is due to the effective dipole moment lowering the ionization energy, with 30% being due to an enhancement in the bonding and other final state

effects. Averaging over all of the clusters studied here, about 79% of effect is due to electrostatic effect, and 21% is due to the binding enhancement. We note that both of these effects are mostly independent of the electronic structure of the cluster core.

**Effect of ligands on PAl$_{12}$**. We now show how the shell and ligand effect can be combined to create a motif with exceptionally low ionization energy. As we mentioned earlier, PAl$_{12}$ has one electron over the filled shell of 40 $e^-$ within jellium model resembling the situation in alkali atoms. As the electron count goes from 40 $e^-$ (CAl$_{12}$) to 41 $e^-$(PAl$_{12}$), the ionization energy drops by 1.53 eV due to shell effect shown in Fig. 2. The AIE further decreases by a large value of 3.1 eV with the addition of 3 EP ligands (ligand effect) creating an alkali-like superatom with AIE of 3.25 eV even less than that of any alkali atom.

**Discussion of ligand effects**. In summary, the present studies offer an alternate strategy to control the ionization characteristics of metallic cores that differs from the conventional approach in which filling of electronic shells leads to lower ionization energy. As we show, the ionization energy of metallic cores can be reduced by attaching ligands that form charge transfer complexes. The local charge transfers act to raise the electronic spectrum in a crystal field like effect that progressively lowers the ionization energy as additional ligands are added. Part of this mechanism is due to a cluster version of the change in work function of a surface due to adding a dipole moment, while the remaining effect is due to binding enhancement in the cation and the stabilization of the charged state of the cluster by the ligands. The ligand effect has several new features. First, the ionization energy can be controlled by changing the number of ligands as the shift in the electronic spectra is due to the coulomb well formed by the local charge exchange. Second, the ligand effect is independent of the occupation of the electronic shell of the cluster and hence allows a pathway to create donor species that can donate multiple electrons. Finally the ligand effect can be combined with shell effect to create multiple donor species with very low ionization energies that can have applications including newly discovered superatom doped semiconductors. We would like to add that the while current work focused only on reducing the AIE by adding donor ligands, an opposite effect, namely an increase in the electron affinity using acceptor ligands is under investigation and will be reported later. We hope that the present findings would stimulate further interest in using ligands for control of oxidation state in clusters.

## Methods

**Theoretical techniques**. First-principles studies were carried out using Amsterdam Density functional (ADF) of codes. The generalized gradient approximation (GGA) as proposed by Perdew, Burke, and Ernzerhof (PBE) is used to incorporate the exchange and correlation effects[33]. The Slater-type orbitals (STO) located at the atomic sites are used to make the atomic wave functions. A linear combination of these atomic orbitals are combined to form the cluster wave functions[34]. A TZ2P basis set and a large frozen electron core was used to ascertain completeness. Trial structures for ligated clusters are obtained by attaching the ligands to previously available structures of Al based clusters. The quasi-Newton method is used to obtain the local minimum for each structure without any symmetry restriction. Relativistic effects are incorporated in this study by using the scalar-relativistic zero-order regular approximation (ZORA)[35,36]. The calculations covered the neutral and cationic species, and several possible spin multiplicities were examined.

**Data availability**. The optimized Cartesian coordinates of the structures generated in this study are available from the corresponding author on reasonable request. The energies, spin magnetic moments, and population analyses are included in the Supplementary Information.

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

## Acknowledgements

We gratefully acknowledge support from the US Air Force Office of Scientific Research for this work. We also thank Prof. Tatsuya Tsukuda for providing the cartesian coordinates of Al$_{13}$(EP)$_4$ clusters.

## Author contributions

V.C. conducted all the calculations. S.N.K, A.C.R, and V.C analyzed the results and wrote the manuscript.

## Additional information

**Competing interests:** The authors declare no competing interests.

