## [Peer Review File · Nature Communications]

Reviewers' comments:

Reviewer #1 (Remarks to the Author):

The manuscript entitled "Inducing Alkali Character in Metallic Cores: Not the Nature's Way" by Chaunan and Khanna is a theoretical investigation of the effect of ligand coordination on the ionization energy of Al₁₃ cluster and its doped relative. Overall, I very much enjoyed this article and recommend its publication in Nature Communications. I would, however, suggest making a few changes listed below.

1. The abstract could be stronger. It feels like it undersells the impact of the work, especially the second half of the paragraph.
2. There are small grammatical errors throughout the manuscript that should be corrected.
3. The caption of Figure 2 should be missing the word "and" before the panel label "(b)".
4. I also feel like Fig 2b is out of order. We have to go through Fig. 3 and 4 before reaching the discussion of Fig 2b. I think the paper would gain from a bit of restructuring.
5. Is it possible to attach more than 3 ligands to the cluster? Is the BE = 0 for the fourth ligand?
6. Is there any rational why the meta position seems to be the lowest energy position for the second EP?

More generally, the idea that ligand coordination changes the electronic structure of complexes (through ligand field effect) is obviously extremely well established in chemistry. The proposed concept that it's the charge of the ligand that is responsible for the observed changes in superatoms is interesting but it also begs the question whether better donors (e.g., phosphines or N-heterocyclic carbenes) would have a bigger effect or not. What about a charged ligand where the charge is not on the atom that binds to the metal?

Reviewer #2 (Remarks to the Author):

This manuscript details impactful insights into the dramatic effects ligands play in modulating electronic properties of superatoms. The use of pyrrolidone ligands for stabilizing metallic Al₁₃ superatoms is novel and has wide relevance to many working in the field of theoretical chemistry as well as researchers designing new materials based on superatom building blocks. The authors use density functional calculations to demonstrate that these pyrrolidone ligands can be used to fine-tune the ionization energies of Al₁₂ superatoms without disturbing the HOMO-LUMO gaps. This study further reinforces the similarities between superatoms and transition metal atoms by proposing traditional crystal field effects on a superatomic scale. This is important for designing superatom assembled materials that can be tailored electronically via rational ligand selection. The strategies described in the manuscript have a broad impact and I look forward to seeing if the authors can successfully achieve the same electronic effects with inverted electronic outcomes such as generating superatoms with high electron affinities.

There are several grammatical errors in the abstract and title that should be addressed prior to publication.

Reviewer #3 (Remarks to the Author):

The authors present the results of a theoretical (DFT) study on the influence of a special type of ligands (C₆H₁₁NO) on the electronic structure and especial the ionization potential of small pure and doped aluminum clusters with 13 atoms.

They observe a strong lowering of ionization energies, which is proportional to the number of ligands attached, and independent of the valence electron count in the cluster (so independent of whether the cluster has a closed shell electronic structure or not).

They especially emphasize that this rather weakly bound ligand does not change the number of valence electrons in the metal cluster, in contrast to other more strongly bound ligands.

In principle this is an interesting result. The basic effect is well known (see below), but the amount of the shift of the ionization energies is, at least at first sight, quite surprising.

Lately metal clusters with strongly bound ligands have been discussed in detail; it is certainly good to show that also weakly bound ligands can have a significant effect on metal cluster properties like the ionization potential.

In the present form, however, the report cannot be recommended for publication.

There are a number of important points the authors should first work on.

1. The authors seem to be surprised that the ionization potential can be changed without changing the number of valence electrons. This is, however, not so surprising as it seems.

The ionization potential of a metal cluster in the simplest approximation is given by the bulk work function plus the cluster charging energy. Both contributions will be changed by ligands.

As is described in most solid state physics text books, about half of the work function of metals is produced by an electrostatic effect, the surface dipole layer. This is the reason why different facets of metals have different work functions. Any absorption of atoms or molecules will change the dipole layer, especially so if the ligand molecules have an electric dipole moment (or develop one at the contact point to the surface), and therefore will change the work function.

This has been known for decades; recently there is a renewed interest due to the need of electrode materials with tunable work function (see for example Ford et al. *CS Nano*, 2014, 8 (9), pp 9173–9180) - a publication which should be cited).

The second influence of the the ligand layer is that any dielectric layer on a metal sphere will lower its charging energy, bringing the cluster ionization energy closer to the bulk work function.

So the effects obtained in the calculations seem to be explainable by rather simple (and well known) models. The authors should definitely discuss what is known for bulk surfaces, and maybe present some small model calculations estimating the amount of ionization energy shift (it could still be that there are some surprises, in the sense that the shift is stronger than expected)

2. The authors also seem surprised that the electronic structure of the metal clusters shifts practically rigidly, so all levels get shifted by about the same energy. This however, not so surprising, as the changed surface dipole layer just changes the depth of the effective single particle potential for the electrons; so the level structure can in fact be expected not to change much.

3. The title of the report is somewhat misleading and should be changed. Why is the reduction of the

work function by ligands not Nature's way? Nature does it all the time, simple oxidization of a metal surface will strongly effect its work function by the change of the surface dipole layer. Also "inducing alkali character" is a bit misleading, as it just means a lowering of the ionization energy, not a change of the character of the metal towards a more free-electron-like one, nor a change of the metal cluster electronic structure to a "super alkali" one, which would be a closed shell structure with an additional electron.

So a title like "strong lowering of metal cluster ionization energy by organic ligands" seems to be more appropriate.

4. If time permits, it could also be very helpful if the authors could do a real calculation for the corresponding bulk surface situation, in order to be able to say how much the bulk work function of aluminum is changed by this kind of ligands (and their density).

In order to strengthen the general importance of the report, the authors also might want to say what they believe could be ultimately achieved by ligating, choosing molecules with even stronger dipoles and larger dielectric influence; could one speculate what could be the lowest possible ionization energy of a ligated metal cluster?

REPLY TO REVIEWERS

We are grateful to all the reviewers for a careful reading of the manuscript and for all the constructive suggestions. We have incorporated all the suggestions in the revised manuscript. In the following we outline the changes we have made in response to reviewer comments.

Reviewer #1 (Remarks to the Author):

The manuscript entitled "Inducing Alkali Character in Metallic Cores: Not the Nature's Way" by Chaunan and Khanna is a theoretical investigation of the effect of ligand coordination on the ionization energy of Al_{13} cluster and its doped relative. Overall, I very much enjoyed this article and recommend its publication in Nature Communications. I would, however, suggest making a few changes listed below.

We thank the reviewer for a careful reading of the paper and suggestions.

1. The abstract could be stronger. It feels like it undersells the impact of the work, especially the second half of the paragraph.

We have rewritten the abstract in order to raise the impact of the work.

2. There are small grammatical errors throughout the manuscript that should be corrected.

We have attempted to correct these.

3. The caption of Figure 2 should be missing the word "and" before the panel label "(b)".

New Figure 2 does not have panel (b). See Figure 2(b) as Figure 5(a) in revised manuscript.

4. I also feel like Fig 2b is out of order. We have to go through Fig. 3 and 4 before reaching the discussion of Fig 2b. I think the paper would gain from a bit of restructuring.

We have considered the suggestion and now Fig 2b stands as Fig 5a after the discussion of Fig. 3 and Fig. 4.

5. Is it possible to attach more than 3 ligands to the cluster? Is the BE = 0 for the fourth ligand?

In order to see that whether 4th ligand bind to the cluster, we have considered the Al_{13} and CAI_{12} clusters having open and closed electronic shell respectively. Our calculations find that 4th

ligand **does not bind** to the metal site of $\text{Al}_{13}(\text{EP})_3$ and $\text{CAI}_{12}(\text{EP})_3$ in their ground state as shown in the figure below. In fact, the binding energy of the 4th ligand to $\text{Al}_{13}(\text{EP})_3$ and $\text{CAI}_{12}(\text{EP})_3$ is 0.39 and 0.37 eV respectively which is significantly low.

6. Is there any rational why the meta position seems to be the lowest energy position for the second EP?

We have extended our discussion of this issue. The discussion is as follows on page 7:

“We find that the *para* position is destabilized by the ligand inducing an electron donor site on the opposite side of the clusters with closed electronic shells such as CAI_{12} and SiAl_{12} .¹³ For the case of open shell Al_{13} , and BAI_{12} , the *para* position is only slightly less stable because the induced active site is half-filled. The *ortho* position is close in energy to the *meta* position in all cases, but is destabilized due to steric interactions.”

More generally, the idea that ligand coordination changes the electronic structure of complexes (through ligand field effect) is obviously extremely well established in chemistry. The proposed concept that it's the charge of the ligand that is responsible for the observed changes in superatoms is interesting but it also begs the question whether better donors (e.g., phosphines or N-heterocyclic carbenes) would have a bigger effect or not. What about a charged ligand where the charge is not on the atom that binds to the metal?

We thank reviewer for this suggestion. In order to see the effect of better donor, we chose the N-heterocyclic carbenes ligated CAI_{12} clusters as shown in the figure below. We do find the similar phenomenon where ionization energies are smaller than the corresponding values in case of EP ligands which suggests stronger effect induced by the better electron donating ligands.

AIP=5.33 eV
 $\Delta Q = -0.39 e^-$

AIP=4.54 eV
 $\Delta Q = -0.69 e^-$

AIP=3.92 eV
 $\Delta Q = -0.98 e^-$

CAI_{12} with carbene ligands

We thank the reviewer for all the suggestions.

Reviewer #2 (Remarks to the Author):

This manuscript details impactful insights into the dramatic effects ligands play in modulating electronic properties of superatoms. The use of pyrrolidone ligands for stabilizing metallic Al₁₃ superatoms is novel and has wide relevance to many working in the field of theoretical chemistry as well as researchers designing new materials based on superatom building blocks. The authors use density functional calculations to demonstrate that these pyrrolidone ligands can be used to fine-tune the ionization energies of Al₁₂ superatoms without disturbing the HOMO-LUMO gaps. This study further reinforces the similarities between superatoms and transition metal atoms by proposing traditional crystal field effects on a superatomic scale. This is important for designing superatom assembled materials that can be tailored electronically via rational ligand selection. The strategies described in the manuscript have a broad impact and I look forward to seeing if the authors can successfully achieve the same electronic effects with inverted electronic outcomes such as generating superatoms with high electron affinities.

We thank the reviewer for a careful reading of the manuscript and suggestions. We are currently into the process of finding the reverse effect as suggested.

There are several grammatical errors in the abstract and title that should be addressed prior to publication.

We have changed the title, and attempted to address the grammatical errors in the abstract and manuscript.

We thank the reviewer for all the suggestions.

Reviewer #3 (Remarks to the Author):

The authors present the results of a theoretical (DFT) study on the influence of a special type of ligands (C₆H₁₁NO) on the electronic structure and especial the ionization potential of small pure and doped aluminum clusters with 13 atoms.

They observe a strong lowering of ionization energies, which is proportional to the number of ligands attached, and independent of the valence electron count in the cluster (so independent of whether the cluster has a closed shell electronic structure or not).

They especially emphasize that this rather weakly bound ligand does not change the number of valence electrons in the metal cluster, in contrast to other more strongly bound ligands.

In principle this is an interesting result. The basic effect is well known (see below), but the amount of the shift of the ionization energies is, at least at first sight, quite surprising. Lately metal clusters with strongly bound ligands have been discussed in detail; it is certainly good to show that also weakly bound ligands can have a significant effect on metal cluster properties like the ionization potential.

In the present form, however, the report cannot be recommended for publication.

There are a number of important points the authors should first work on.

We thank the reviewer for a careful reading of the manuscript, and valuable suggestions.

1. The authors seem to be surprised that the ionization potential can be changed without changing the number of valence electrons. This is, however, not so surprising as it seems. The ionization potential of a metal cluster in the simplest approximation is given by the bulk work function plus the cluster charging energy. Both contributions will be changed by ligands. As is described in most solid state physics text books, about half of the work function of metals is produced by an electrostatic effect, the surface dipole layer. This is the reason why different facets of metals have different work functions. Any absorption of atoms or molecules will change the dipole layer, especially so if the ligand molecules have an electric dipole moment (or develop one at the contact point to the surface), and therefore will change the work function. This has been known for decades; recently there is a renewed interest due to the need of electrode materials with tunable work function (see for example Ford et al. CS Nano, 2014, 8 (9), pp 9173–9180) - a publication which should be cited).

The second influence of the the ligand layer is that any dielectric layer on a metal sphere will lower its charging energy, bringing the cluster ionization energy closer to the bulk work function. So the effects obtained in the calculations seem to be explainable by rather simple (and well known) models. The authors should definitely discuss what is known for bulk surfaces, and maybe present some small model calculations estimating the amount of ionization energy shift (it could still be that there are some surprises, in the sense that the shift is stronger than expected)

We have expanded our discussion of the effect of the ligand on the ionization potential and included the analogy with the band bending effect observed on surfaces. For comparison, we show below the effect of the dipole moment of EP on the Al(111), and Al(100) surfaces. The shift in the work function corresponds to 1.63 eV on Al(111), and 1.77 eV on Al(100). Note that the different densities lead to different changes in the work function.

There is a reasonable analogy between the surface dipole effect on the work function, and the effect we are seeing here. To better understand this, we have analyzed the shift in the HOMO levels of the clusters, essentially a pure dipole effect due to the surface ligands and the actual calculated change in the ionization potentials. We find that around 70 % of the shift is due to the dipole, with the remaining stabilization being due to final state effects, either an enhancement of the binding of the ligand due to the change in the charge state of the cluster, or perhaps the dielectric effect of the ligand. The expanded discussion is copied below for clarity.

“Therefore, a dominating effect of ligands is to create a crystal field like effect that raises the electronic spectrum while making the charge transfer complexes. Part of this effect is analogous to the change in work function caused by a dipole moment on a surface.^{25–28,32} A second related effect is that the binding energy of the ligand to the charged cluster is likely to be larger than to a neutral cluster. To separate these effects, we may use the absolute position of the HOMO in the neutral cluster to indicate the initial state electrostatic lowering of the energy levels, and the change in the adiabatic ionization potential to include both electrostatic effects, the binding enhancement of the ligand, and any other final state stabilization of the cation that may be caused by the ligands. The change in the absolute position of the HOMO of the neutral cluster is analogous to the relative position of the vacuum energy and Fermi energy in a surface. In our method, the vacuum level is 0 eV by definition, so the absolute position of the HOMO includes this electrostatic effect. For example, the addition of EP to Al₁₃ lowers the HOMO by 0.70 eV, while the AIP is lowered by 1.05 eV. This suggests that about 70% of the effect is due to the effective dipole moment lowering the ionization potential, with 30 % being due to an enhancement in the bonding and other final state effects. Averaging over all of the clusters studied here, about 79% of effect is due to electrostatic effect, and 21% is due to the binding enhancement. We note that both of these effects are mostly independent of the electronic structure of the cluster core.”

2. The authors also seem surprised that the electronic structure of the metal clusters shifts practically rigidly, so all levels get shifted by about the same energy. This however, not so surprising, as the changed surface dipole layer just changes the depth of the effective single particle potential for the electrons; so the level structure can in fact be expected not to change much.

Agreed, see answer to 1.

3. The title of the report is somewhat misleading and should be changed. Why is the reduction of the work function by ligands not Nature's way? Nature does it all the time, simple oxidization of a metal surface will strongly effect its work function by the change of the surface dipole layer. Also "inducing alkali character" is a bit misleading, as it just means a lowering of the ionization energy, not a change of the character of the metal towards a more free-electron-like one, nor a change of the metal cluster electronic structure to a "super alkali" one, which would be a closed shell structure with an additional electron.

So a title like "strong lowering of metal cluster ionization energy by organic ligands" seems to be more appropriate.

Following the reviewer's suggestion, we have changed the title to

"Strong Lowering of Ionization Energy in Metallic Clusters via Ligands: Beyond Electronic Shell Filling"

We thank the reviewer for the suggestion.

4. If time permits, it could also be very helpful if the authors could do a real calculation for the corresponding bulk surface situation, in order to be able to say how much the bulk work function of aluminum is changed by this kind of ligands (and their density).

In order to strengthen the general importance of the report, the authors also might want to say what they believe could be ultimately achieved by ligating, choosing molecules with even stronger dipoles and larger dielectric influence; could one speculate what could be the lowest possible ionization energy of a ligated metal cluster?

A calculation of the work functions is shown above.

REVIEWERS' COMMENTS:

Reviewer #1 (Remarks to the Author):

The authors addressed all my comments. I recommend publication in Nature Communications.

Reviewer #3 (Remarks to the Author):

The authors have responded adequately to the points raised.

The only suggestion (or wish) I still have is that they add the interesting AI surface calculation results to the supporting informations, as other readers might also be interested in them.

Otherwise the report can now be recommended for publication.

(there still are grammatical errors, but I guess some additional editing will be done by Nature Communications)

REPLY TO REVIEWERS

We are grateful to all the reviewers for a careful reading of the manuscript and for all the constructive suggestions. We have incorporated all the suggestions in the revised manuscript. In the following we outline the changes we have made in response to reviewer comments.

Reviewer #3 (Remarks to the Author):

The authors have responded adequately to the points raised.

The only suggestion (or wish) I still have is that they add the interesting

AI surface calculation results to the supporting informations, as other readers might also be interested in them.

Otherwise the report can now be recommended for publication.

We have added the information on AI surface calculation in supporting information as desired by the reviewer. We have also added a sentence on page 11 pointing to this information.

We are truly grateful to this reviewer for his/her comments.